# Referring Human Pose and Mask Estimation in the Wild

**Bo Miao**[1]    **Mingtao Feng**[2]    **Zijie Wu**[3]    **Mohammed Bennamoun**[1]
**Yongsheng Gao**[4]    **Ajmal Mian**[1]

[1]University of Western Australia  [2]Xidian University  [3]Hunan University  [4]Griffith University

https://github.com/bo-miao/RefHuman

## Abstract

We introduce Referring Human Pose and Mask Estimation (R-HPM) in the wild, where either a text or positional prompt specifies the person of interest in an image. This new task holds significant potential for human-centric applications such as assistive robotics and sports analysis. In contrast to previous works, R-HPM (i) ensures high-quality, identity-aware results corresponding to the referred person, and (ii) simultaneously predicts human pose and mask for a comprehensive representation. To achieve this, we introduce a large-scale dataset named RefHuman, which substantially extends the MS COCO dataset with additional text and positional prompt annotations. RefHuman includes over 50,000 annotated instances in the wild, each equipped with keypoint, mask, and prompt annotations. To enable prompt-conditioned estimation, we propose the first end-to-end promptable approach named UniPHD for R-HPM. UniPHD extracts multimodal representations and employs a proposed pose-centric hierarchical decoder to process (text or positional) instance queries and keypoint queries, producing results specific to the referred person. Extensive experiments demonstrate that UniPHD produces quality results based on user-friendly prompts and achieves top-tier performance on RefHuman `val` and MS COCO `val2017`.

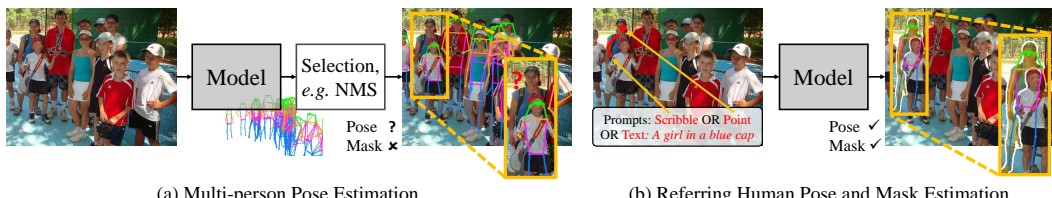

(a) Multi-person Pose Estimation                    (b) Referring Human Pose and Mask Estimation

Figure 1: Task illustration of (a) multi-person pose estimation predicts numerous outcomes and requires selection strategies during deployment, potentially leading to false negatives or suboptimal target results, and (b) our referring human pose and mask estimation requires a *unified promptable* model to simultaneously predict accurate pose and mask for the person of interest, providing *comprehensive and identity-aware* human representations to benefit human-AI interaction.

## 1   Introduction

Human pose estimation in the wild is a fundamental yet challenging problem in the vision community, fueling advancements in various applications like human-AI interaction, activity analysis, video surveillance, assistive robotics, and sports analysis. This task aims to locate keypoints (joint locations) of humans within images in unconstrained environments.

38th Conference on Neural Information Processing Systems (NeurIPS 2024).

Previous multi-person pose estimation techniques typically follow a two-stage paradigm, separating the problem into person detection and local keypoint regression. These techniques can be summarized as top-down and bottom-up approaches. Top-down approaches [9, 15, 63, 79, 84] use a detection model to identify human bounding boxes and a separate pose estimation model to predict keypoints on the cropped single-human image. The independent models in these methods lead to a non-end-to-end pipeline with substantial computational costs. Bottom-up approaches [3, 10, 31, 62] usually predict keypoint heatmaps to obtain instance-agnostic keypoints and assign them to individuals using heuristic grouping algorithms. The intricate grouping algorithms in these approaches introduce hand-designed parameters and face challenges in handling complex scenarios such as occlusions.

With advancements in attention mechanisms [2, 56, 57, 59, 77] and transformer architectures [4, 95], many recent approaches regard human pose estimation as a direct set prediction problem and design end-to-end differentiable transformers, leading to notable improvements. They employ bipartite matching to establish one-to-one instance correspondence for the set of predictions, avoiding the need for post-processing in the training stage. Among them, PETR [70] proposes a transformer that progressively predicts keypoint positions through pose and keypoint decoders. QueryPose [85], ED-Pose [87], and GroupPose [44] further incorporate a human detection stage for instance feature extraction or query initialization to improve performance and expedite model convergence. They all use keypoint-level queries to capture local details for accurate pose estimation.

Despite demonstrating favorable performance, these methods still require designed strategies to select the best match for a target individual during deployment, which can result in suboptimal outcomes or false negatives, and lack exploration of human-AI interaction to directly predict expected results based on natural prompts. Additionally, they overlook joint human pose and mask estimation, which provides comprehensive human representations to facilitate applications like assistive robotics and sports analysis. For example, accurate joint human pose and mask estimation in unconstrained environments enables robots to locate, analyze, and interact with target individuals, enhancing user experience and assistive tasks.

In this paper, we propose the new task of Referring Human Pose and Mask Estimation (R-HPM) in the wild. As illustrated in Figure 1, unlike multi-person pose estimation [3, 15, 24, 66], R-HPM is a multimodal reasoning task that requires a *unified* model to predict both pose and mask of a referred individual using user-friendly text or positional prompts, enabling comprehensive and identity-aware human representations without post-processing. To achieve this, we introduce RefHuman, a large-scale dataset that substantially extends MS COCO [40] with additional text, scribble, and point prompt annotations. Our dataset accommodates diverse task settings to enhance human-AI interaction. Manually annotating text descriptions and scribbles is expensive. To reduce annotation costs, we design a human-in-the-loop text generation strategy using powerful large language models and employ bezier curves to automate scribble generation.

To benchmark R-HPM, we propose an end-to-end promptable approach called UniPHD. Our approach directly adopts prompts as instance queries and coordinates them with keypoint queries to jointly predict identity-aware keypoint positions and masks for the target. Unlike previous works [44, 70, 85, 87], UniPHD introduces a pose-centric hierarchical decoder (PHD) that employs deformable and graph attention to effectively model local details and global dependencies, ensuring target-awareness and instance coherence. Furthermore, our approach follows a general paradigm that enables seamless integration with the decoders from recent transformer-based methods such as GroupPose [44] and ED-Pose [87] for R-HPM. We conduct extensive experiments on the RefHuman dataset. Integrating our approach with GroupPose and ED-Pose yields promising results in an end-to-end fashion. Moreover, our UniPHD approach achieves top-tier performance on both RefHuman `val` and MS COCO `val2017`, demonstrating the effectiveness of our approach and the significance of our task.

Our main contributions are summarized below:

- We propose Referring Human Pose and Mask Estimation (R-HPM) in the wild, a new task that simultaneously predicts the pose and mask of a specified individual using natural, user-friendly text or positional prompts. This task enhances models with identity-awareness and produces comprehensive human representations to benefit human-AI interaction.

- We introduce RefHuman, a large-scale dataset that substantially extends COCO for R-HPM. RefHuman contains pose and mask annotations for individuals in diverse, unconstrained environments and is enriched with corresponding text and positional prompts.

- We propose an end-to-end promptable UniPHD approach for R-HPM. Our approach performs pose-centric hierarchical decoding, achieving top-ranked performance and establishing a solid benchmark for future advancements in this field.

## 2   Related Work

### 2.1   2D Human Pose Estimation in the Wild

Human pose estimation aims to localize keypoints of individuals in unconstrained environments. For a long period, two-stage approaches have dominated this field, generally divided into top-down and bottom-up methods. Top-down methods [9, 15, 34, 63, 79, 84] first detect and crop each person using an object detector, then perform pose estimation on these cropped instance-level images using a separate model. Although effective, the redundancy from the extra detection step, region of interest operations, and separate training makes them suboptimal. Bottom-Up methods [3, 10, 31, 62] first detect abundant instance-agnostic keypoints and then group them into individual poses. While generally efficient, their intricate grouping algorithms pose challenges in handling complex scenarios, resulting in inferior performance. Furthermore, these two-stage approaches suffer from non-differentiable, hand-crafted post-processing steps that challenge optimization. Inspired by the one-stage object detectors [21, 76], pixel-wise regression methods [49, 53, 55, 64, 73, 75, 78, 81, 93] densely predict pose candidates in an end-to-end fashion and apply Non-maximum Suppression (NMS) to obtain poses for different individuals. However, these methods produce redundant results, challenging the removal of duplicates.

Recent human pose estimation methods [51, 52, 72] explore transformer-based architectures [2, 4, 77, 95] due to their sparse, end-to-end design and promising performance. These methods treat human pose estimation as a direct set prediction problem and use bipartite matching to establish one-to-one instance correspondence during training. Among these, [70] proposes a transformer with a pose decoder to predict keypoints and a joint decoder to refine them. [85] performs estimation on extracted object features to reduce noisy context. [85, 87] incorporate a human detection stage for query initialization to enhance performance. These methods achieve favorable performance but require complex strategies to identify the best match for a specified person during deployment due to the lack of exploration of prompt reasoning. In this work, we propose a multimodal reasoning task to directly and jointly predict the identity-aware pose and mask for a referred person, resulting in comprehensive human body representations and facilitating applications in human-AI interaction.

### 2.2   Referring Image Segmentation

Referring image segmentation (RIS) aims to segment target objects in images based on natural linguistic descriptions [18, 30, 91]. It related tasks include interactive image segmentation [7, 8, 25, 37, 41, 86], which segments targets based on user clicks. Early RIS methods [6, 17, 18, 35, 43, 54, 71, 90] employ convolution and recurrent neural networks for multimodal encoding and segmentation. Their intrinsic constraints in capturing long-range dependencies and handling free-form features often lead to suboptimal performance. To improve multimodal representation and alignment, attention-based bidirectional [19, 20] and progressive [16, 22, 23, 88, 89] cross-modal interaction modules are proposed. Additionally, [80, 92] leverage the strong cross-modal alignment capabilities of pre-trained large language models [67, 69].

With advancements in transformer architectures [4, 61, 77, 95], [11, 13, 29, 38, 42, 45, 58, 60, 83, 94] design end-to-end transformers for language-conditioned segmentation and achieve favorable performance. In this work, we investigate a unified promptable transformer that effectively processes both text and positional prompts, thereby broadening its generalizability and application scope. Moreover, our model coordinates prompts with keypoint queries, enabling the joint prediction of keypoint positions and segmentation masks for specified individuals.

## 3   RefHuman Dataset

We substantially extend COCO [40] to construct the RefHuman dataset. It contains pose and mask annotations for humans along with text and positional prompts to facilitate the new task of R-HPM.

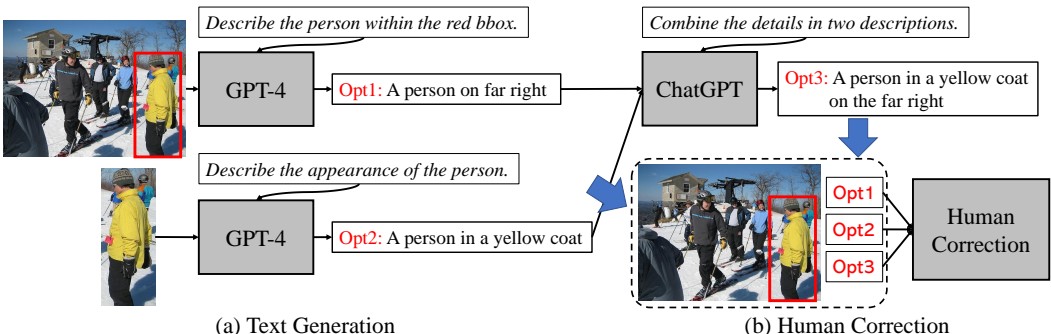

(a) Text Generation  (b) Human Correction

Figure 2: Human-in-the-loop text prompt generation. We use GPT to generate descriptions with complementary local details and global context, then manually review/correct the descriptions.

## 3.1 Data Annotation

Pioneer works on 2D human pose estimation datasets in RGB images include [5, 12, 14, 26, 28]. Recent efforts focus on human pose estimation 'in the wild', establishing datasets such as MPII [1], MS COCO [40], CrowdPose [33], COCO-WholeBody [27], and AI Challenger [82]. Despite their prevalence, these datasets lack crucial text and scribble prompt annotations needed for human-AI interaction. The large-scale RefHuman dataset extends the widely-used MS COCO with additional text and positional (scribbles and points) prompt annotations, facilitating promptable models to enhance human-AI interaction.

**Text Prompt Annotation.** Manually annotating text descriptions is costly. To mitigate this, we design a human-in-the-loop text generation strategy using the large language model, ChatGPT [65]. As shown in Figure 2 (a), we first provide the entire image with a bounding box indication to GPT-4 to generate a text description [Opt1] with global context. However, large language models often misidentify targets in complex scenes. To handle this, we crop the image to focus on the target person and let GPT-4 generate a description [Opt2] with correct local details. Given that individuals may have similar appearances, descriptions generated based on instance-level images are often not distinguishable. Therefore, we combine the global and local descriptions, integrating them into a comprehensive description [Opt3] of the target person. Finally, we manually select and revise these automatically generated descriptions to create accurate text prompts.

Our strategy successfully generates acceptable descriptions for images with simple scenarios. These correspond to over 35% of the cases. In the remaining cases, unsatisfactory descriptions are generated. Common issues include misidentification of individuals, incorrect orientation (*e.g.*, facing left or right), and indistinguishable context. Nevertheless, these descriptions often provide valuable context, expediting the annotation process. To further scale up the RefHuman training set, we integrate text annotations from RefCOCO/+/g [30, 50], doubling the number of referred instances.

**Positional Prompt Annotation.** Bézier curves are parametric curves based on Bernstein Polynomials and widely used in computer graphics. In this work, we employ cubic Bézier curves to simulate scribbles and generate clicks. Given four control points $\mathbf{p_0}$, $\mathbf{p_1}$, $\mathbf{p_2}$, and $\mathbf{p_3}$, the cubic Bézier curve starts at $\mathbf{p_0}$ moving toward $\mathbf{p_1}$ and reaches $\mathbf{p_3}$ from the direction of $\mathbf{p_2}$. The general equation for a Bézier curve $\mathbf{c}(t, n)$ of degree $n$ is:

$$\mathbf{c}(t, n) = \sum_{i=0}^{n} b_{i,n}(t)\mathbf{p}_i, \quad 0 \leq t \leq 1 \tag{1}$$

$$b_{i,n}(t) = \binom{n}{i}(1-t)^{n-i}t^i, \quad i = 0, ..., n \tag{2}$$

where $b_{i,n}(t)$ represents the Bernstein basis polynomials and $\binom{n}{i}$ is the binomial coefficient. To generate scribble prompts, we randomly sample four control points within the foreground mask to fit a cubic Bézier curve, represented as an ordered set of points $\{(x_1, y_1), (x_2, y_2), ..., (x_n, y_n)\}$. The curve is then discretized by uniformly sampling twelve points to form a scribble prompt $\mathbf{s} = \{(x_{\lfloor \frac{kn}{12} \rfloor}, y_{\lfloor \frac{kn}{12} \rfloor}) \mid k = 1, 2, ..., 12\}$. For point prompts, we randomly select a single point $\mathbf{p} = (x, y)$ from $\mathbf{s}$, where $x$ and $y$ are the horizontal and vertical coordinates.

Table 1: Data statistics of human-related images in RefCOCO [30], RefCOCO+ [30], RefCOCOg [50], their combined dataset RefCOCO/+/g, and our RefHuman.

| Dataset | Prompt | Target | Image | Instance | Expression |
|---|---|---|---|---|---|
| RefCOCO | Text | Mask | 9209 | 22819 | 65550 |
| RefCOCO+ | Text | Mask | 9209 | 22804 | 66463 |
| RefCOCOg | Text | Mask | 9141 | 16435 | 32299 |
| RefCOCO/+/g | Text | Mask | 13468 | 29470 | 149278 |
| **RefHuman (Ours)** | **Text, Scribble, Point** | **Pose, Mask** | **21634** | **50210** | **170017** |

## 3.2 Data Statistics

As shown in Table 1, RefHuman includes 50,210 human instances across 21,634 images, larger than RefCOCO [30], RefCOCO+ [30], and RefCOCOg [50], and with additional positional prompts. To construct RefHuman `train` set, we annotate prompts for all humans in MS COCO `train2017` set with at least three surrounding people, a minimum of eight visible keypoints, and an area ratio of at least 2%. For the RefHuman `val` set, we annotate humans in MS COCO `val2017` set, excluding those with non-visible keypoints or an area ratio below 1%, as instances below this threshold are often not visually clear and difficult to describe accurately. Note that each instance may have multiple text, scribble, and point annotations, with each image-prompt pair treated as a separate sample.

## 4 Method

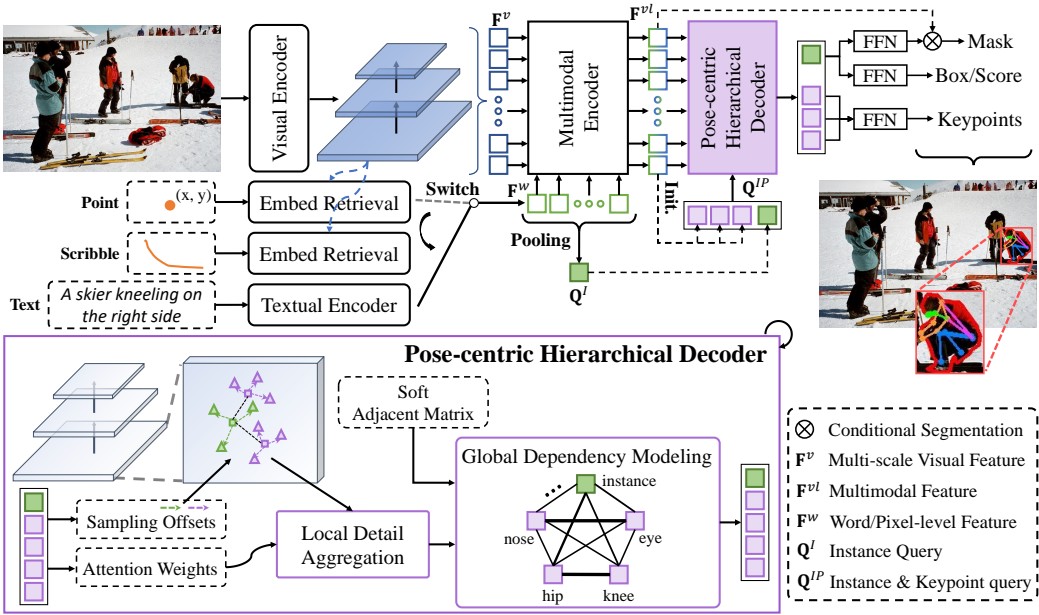

Figure 3: Detailed architecture of our UniPHD, which contains a multimodal encoder that imbues visual features with prompt awareness and a pose-centric hierarchical decoder that enables prompt-conditioned queries to effectively capture local details and global dependencies within targets. Our unified model is end-to-end and accepts text descriptions, scribbles, or points as prompts to predict the keypoint positions and segmentation mask of the target person.

Given an image $\mathbf{I}$ and a text prompt $\mathbf{T}$ or positional prompt $\mathbf{P}$, our task aims to simultaneously predict the keypoint positions $\mathbf{K} \in \mathbb{R}^{N \times 2}$ and binary segmentation mask $\mathbf{M} \in \mathbb{R}^{H \times W}$ of the referred person, where $N$ is the number of keypoints, and $H$ and $W$ are the spatial dimensions. To this end, we propose a fully end-to-end promptable UniPHD approach, as illustrated in Figure 3. UniPHD consists of four key components: Backbone, Multimodal Encoder, Pose-centric Hierarchical Decoder, and Task-specific Prediction Heads. A small set of prompt-conditioned queries is employed to identify

the referred person and estimate results. During inference, we directly output expected predictions from the highest-scoring query group without any post-processing.

## 4.1 Backbone

**Visual Encoder.** We start by using a visual encoder to extract multi-scale visual features and apply a point-wise convolution to reduce the channel dimension of features to $D = 256$ for efficient multimodal interactions. Feature maps with downsampling rates of $\{8, 16, 32, 64\}$ are then flattened and concatenated into tokenized visual representations $\mathbf{F}^v$. We adopt Swin Transformer [47] as the visual encoder in this work.

**Prompt Encoder.** For a linguistic description with $L$ words as a prompt, we use RoBERTa [46] to extract word-level features $\mathbf{F}^w \in \mathbb{R}^{L \times D}$ and generate sentence-level feature as the instance query $\mathbf{Q}^I \in \mathbb{R}^{1 \times D}$ by pooling $\mathbf{F}^w$. For a discretized scribble or a single point prompt, we retrieve embeddings directly from visual features at stride 16 based on their positions to obtain pixel-level prompt features $\mathbf{F}^w$, which are then pooled to form the instance query $\mathbf{Q}^I$.

## 4.2 Multimodal Encoder

To generate target-related multimodal representations $\mathbf{F}^{vl}$, visual tokens $\mathbf{F}^v$ are first enhanced through a *modality-specific* cross-attention, which integrate information from prompt features:

$$\mathbf{F}^{vl} = \mathbf{F}^v + \text{Attn}_i(Q = \mathbf{F}^v, K = \mathbf{F}^w, V = \mathbf{F}^w) \tag{3}$$

where $i = 0$ for text prompt and $i = 1$ for positional prompts, *i.e.*, we adopt separate parameters for cross-modal fusion of different prompt types. After that, we follow previous works [44, 58, 87] to use a memory-efficient deformable transformer encoder to encode the multimodal features. The output of the transformer encoder $\mathbf{F}^{vl}$ is then forwarded to the decoder to update queries.

## 4.3 Pose-centric Hierarchical Decoder

We employ $n$ groups of queries for the referred human and generate $n$ groups of results. Each group consists of a prompt-conditioned instance query and $k$ learnable keypoint queries. The keypoint queries regard each keypoint as a target and aim to regress their respective positions, while each instance query, conditioned on the prompt, predicts a confidence score, dynamic segmentation filters, and a bounding box to locate the referred human. The scores are supervised by the losses of each query group to indicate result quality. During deployment, we directly generate results for the target using the highest-scoring query group without any post-processing.

**Prompt-conditioned Query Initialization.** We first construct a query group template $\mathbf{Q}^{IP}_{Init} \in \mathbb{R}^{(k+1) \times D}$ by concatenating the text/positional prompt embedding $\mathbf{Q}^I$ with $k$ (*e.g.*, 17) randomly initialized learnable embeddings $\mathbf{Q}^P$. To generate $n$ query groups, we apply linear layers to the prompt-aware multimodal features $\mathbf{F}^{vl}$ to identify the top-$n$ highest-scoring positions $\mathbf{p}$ and estimate their corresponding keypoint positions and centers as reference points, denoted as $\mathbf{c} \in \mathbb{R}^{n \times (k+1) \times 2}$. The template $\mathbf{Q}^{IP}_{Init}$ is then repeated $n$ times and enhanced with reference points $\mathbf{c}$, their associated positional embeddings, and the top-$n$ highest-scoring multimodal embeddings $\mathbf{F}^{vl}_{\mathbf{p}}$, to form $\mathbf{Q}^{IP}$. Finally, the instance query in each copy incorporates the prompt embedding and a positional embedding derived from the keypoints center of each candidate for pose-centric decoding. During decoding process, we employ graph-attention to model global dependencies for each query group, enabling all queries to utilize the prompt as guidance to identify the referred person only.

**Hierarchical Local Context and Global Dependency Modeling.** We introduce a pose-centric hierarchical decoder to effectively model complementary local details and global dependencies for target-aware decoding. As illustrated in Figure 3, local details are first aggregated using the efficient deformable attention [95], similar to [44, 58, 87]. Each instance and keypoint query in $\mathbf{Q}^{IP}$ independently predicts its sampling offsets and corresponding attention weights on the multimodal features $\mathbf{F}^{vl}$. We then aggregate the sampled features accordingly for each query to capture local details. However, after local detail aggregation, the keypoint queries struggle to perceive the prompt information and lacks interactions with each other, challenging the target-awareness and instance coherence. Additionally, the instance query lacks sufficient relevant global context to accurately locate the referred person at pixel-level.

To address these issues, we model each query group as a bipartite graph $\mathbf{G} = \{\mathbf{V}, \mathbf{E}, \mathbf{A}\}$, with nodes $\mathbf{V}$ representing different queries and edges $\mathbf{E}$ denoting relations between nodes, constrained by a learnable soft adjacent matrix $\mathbf{A} \in \mathbb{R}^{(k+1) \times (k+1)}$, which simulates *inherent* keypoint-to-keypoint, keypoint-to-instance, and instance-to-keypoint relations. The edge $\mathbf{E}_{ij}$ from the $i$-th node to the $j$-th node can be formulated as:

$$\mathbf{E}_{ij} = (W^q \mathbf{V}_i)(W^k \mathbf{V}_j)^\top + \mathbf{A}_{ij} \tag{4}$$

where $W^q$ and $W^k$ are learnable projection matrices. Through our graph attention, both instance and keypoint queries capture global dependencies of the target and thus ensure instance coherence. Ultimately, the instance query generates dynamic filters for mask prediction, while the keypoint queries regress target keypoint positions.

### 4.4 Task-specific Prediction Heads

As illustrate in Figure 3, four lightweight heads are built on top of the decoded queries to predict bounding boxes, class scores, masks, and keypoint positions for the target. The box head predicts the bounding box location of the target to aid the learning process. The class head outputs confidence scores, supervised by losses from each query group, to indicate the prediction quality of each query group. The mask head produces dynamic filters for conditional segmentation on the multimodal features $\mathbf{F}^{vl}$. The keypoint head predicts keypoint positions and their visibility for the referred person in images. Detailed training loss functions for these outputs are supplied in the Appendix.

**Conditional Segmentation.** After extracting semantically-rich multimodal features $\mathbf{F}^{vl}$, we address pixel-level target localization using dynamic filters generated by instance queries, which capture both local details and global dependencies. Similar to [58], we standardize the resolution of the multi-scale multimodal features $\mathbf{F}^{vl}$ to $H/8 \times W/8$ and combine them into a single feature map $\mathbf{F}^m$ through efficient element-wise addition. We then reshape the prompt-conditioned dynamic filters to form two point-wise convolutions that apply to $\mathbf{F}^m$ to obtain the segmentation mask for the referred person.

## 5 Experiments

**Metrics.** We use standard metrics to evaluate our task. For pose estimation, we adopt OKS-based average precision (AP) and PCKh at a 0.5 threshold (PCKh@0.5) [1]. For segmentation, we report mask AP and overall Intersection-over-Union (IoU). The AP metrics are evaluated across all query groups, consistent with prior works [44, 87], while PCKh@0.5 and oIoU are measured using only the highest-scoring query group to better reflect real-world deployment.

**Implementation Details.** We train our model on the RefHuman `train`, which contains approximately 46K person instances with 17 keypoints per instance, and evaluate on the RefHuman `val`. We also evaluate our model on MS COCO by generating positional prompts for *all* images in the dataset. Due to the page limit, we leave the further implementation details in the Appendix.

### 5.1 Main Results

In Table 2, we evaluate our models and the recent advances [44, 87] on the RefHuman `val` set. For fairness, all models are evaluated with inputs resized to a maximum of 640 pixels on the longer side.

**Effectiveness of Our Promptable Paradigm.** Recent human pose estimation methods like Group-Pose [87] and ED-Pose [44] predict poses for multiple humans but require hand engineered selection strategies to identify the best match for a specified person, which can result in suboptimal outcomes or false negatives. By integrating their decoders into our end-to-end promptable paradigm, we directly generate poses and masks in one go for the referred person, achieving up to **73.0** pose AP, **89.6** PCKh@0.5, and **85.6** oIoU with scribble prompts. This performance rivals that of previous models which use 3× more training data and hand engineered result selection strategies. Intersection-based result selection is chosen in Table 2 because distance- and IoU-based strategies lead to inferior performance, as discussed in the ablation study. Furthermore, our paradigm achieves advanced performance even with a single point prompt, while previous models struggle with such simple guidance, as the random positive points/clicks can be near poses of unintended humans in crowds. These results demonstrate the effectiveness of our end-to-end promptable paradigm and the significance of our proposed R-HPM task, *i.e.*, accurate joint pose and mask estimation based on user-friendly prompts.

Table 2: Results on RefHuman `val` split. Uni-ED-Pose and Uni-GroupPose integrate ED-Pose [87] and GroupPose [44] into our end-to-end paradigm. *Intersection-based result selection*: selects results covering at least 30% of the ground truth box. [†]: trains models using complete images in COCO `train2017`. FPS is measured on RTX 3090 with a batch size of 24. Uni-ED-Pose and Uni-GroupPose are trained with less data but rival the performance of vanilla models, which perform post-processing with ground truth boxes. Our UniPHD approach achieves top-tier performance.

| | Prompt | Backbone | Pose Estimation | | Segmentation | | | |
| --- | --- | --- | --- | --- | --- | --- | --- | --- |
| | | | AP | PCKh@0.5 | oIoU | AP | Params | FPS |
| *with intersection-based result selection using ground truth boxes* | | | | | | | | |
| GroupPose[†] [44] | BBox | Swin-T | 72.0 | - | - | - | - | - |
| ED-Pose[†] [87] | BBox | Swin-L | 72.8 | - | - | - | - | - |
| GroupPose[†] [44] | BBox | Swin-L | 73.4 | - | - | - | - | - |
| *our R-HPM methods supporting **various** prompts, without result selection strategy* | | | | | | | | |
| Uni-ED-Pose | Text | Swin-T | 65.3 | 78.3 | 74.5 | 61.5 | - | - |
| Uni-GroupPose | Text | Swin-T | 65.0 | 78.0 | 74.7 | 61.4 | - | - |
| **UniPHD** | Text | Swin-T | **66.7** | **79.0** | **75.0** | **62.2** | - | - |
| Uni-ED-Pose | Point | Swin-T | 71.9 | **88.9** | 82.8 | 67.7 | - | - |
| Uni-GroupPose | Point | Swin-T | 71.5 | 88.5 | 82.2 | 67.6 | - | - |
| **UniPHD** | Point | Swin-T | **73.5** | 88.7 | **83.1** | **68.9** | - | - |
| Uni-ED-Pose | Scribble | Swin-T | 72.9 | 89.6 | 84.9 | 69.2 | 175.7M | 35.4 |
| Uni-GroupPose | Scribble | Swin-T | 73.0 | 89.3 | 85.6 | 69.0 | 177.7M | 35.6 |
| **UniPHD** | Scribble | Swin-T | **74.7** | **90.4** | **85.7** | **70.0** | 184.0M | 35.3 |

Table 3: Comparison with state-of-the-art methods on MS COCO `val2017`. Our method achieves leading performance in pose estimation while also offering segmentation capabilities.

| | Max Res. | Backbone | Pose AP | $AP_{50}$ | $AP_{75}$ | $AP_M$ | $AP_L$ |
| --- | --- | --- | --- | --- | --- | --- | --- |
| PETR [70] | 640 | Swin-L | 66.4 | 88.1 | 73.0 | 56.5 | 80.0 |
| ED-Pose [87] | 640 | Swin-L | 68.9 | 89.8 | 75.3 | 60.1 | 81.1 |
| GroupPose [44] | 640 | Swin-T | 68.5 | 88.9 | 75.4 | 60.5 | 79.9 |
| **UniPHD** w/ Point | 640 | Swin-T | 73.5 | 93.7 | 81.1 | 68.0 | 81.8 |
| **UniPHD** w/ Scribble | 640 | Swin-T | **73.9** | **93.9** | **81.9** | **68.5** | **82.2** |

**Effectiveness of Our UniPHD Approach.** In Table 2, our UniPHD approach sets a new benchmark in overall performance for R-HPM through pose-centric hierarchical decoding, outperforming the nearest competitor, Uni-GroupPose w/ Scribble, by **1.7** Pose AP and **1.0** Mask AP, with comparable FPS. Intuitively, the interdependent influence of human keypoints makes graph networks suitable for modeling their dependencies. After applying deformable attention to capture local details, we employ graph attention with a learnable soft adjacent matrix to simulate keypoint-to-keypoint, keypoint-to-instance, and instance-to-keypoint relations. This matrix guides edge construction to effectively model global dependencies and ensure instance-awareness and coherence.

**Results for Different Prompts.** Table 2 shows that scribble prompts outperform point and text prompts in R-HPM. Scribble prompts offer more explicit positional guidance, enhancing instance query robustness. Point prompts, while slightly less accurate, offer greater user convenience in human-AI interaction due to their single-click simplicity. Text prompts face challenges in multimodal alignment, especially for crowded scenarios, but provide linguistic flexibility and enable non-physical interactions. Overall, our unified model effectively handles various prompts, showing great potential to facilitate human-AI interaction.

**Evaluation on MS COCO `val2017`.** We further evaluate UniPHD on MS COCO by aggregating results from all instances in each image. As shown in Table 3, all models are evaluated using inputs resized to a maximum of 640 pixels on the longest side for comparison. Trained solely with positional prompts, UniPHD achieves state-of-the-art performance on COCO `val2017`, even when compared

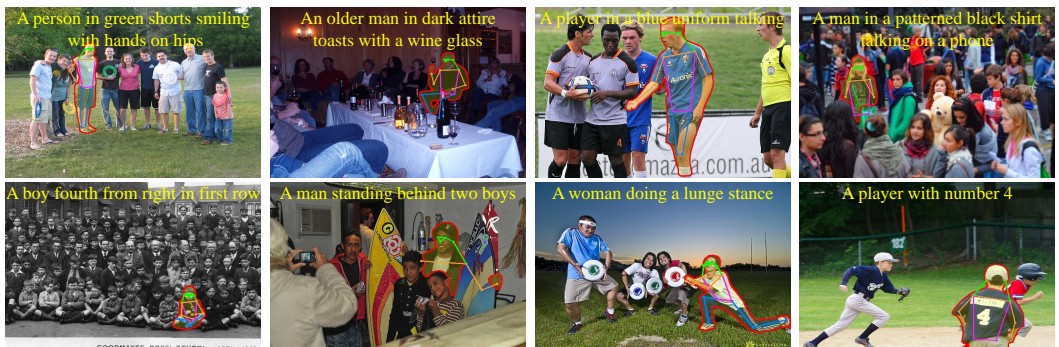

Figure 4: Qualitative results of our UniPHD with text prompts in various challenging scenarios.

Table 4: Comparison with state-of-the-art text-based segmentation methods on RefHuman.

|  | Backbone | oIoU |
|---|---|---|
| LAVT [88] | Swin-T | 74.5 |
| CGFormer [74] | Swin-T | 75.3 |
| ReLA [42] | Swin-T | 75.9 |
| SgMg [58] | Swin-T | 75.9 |
| **Ours** | Swin-T | **76.3** |

Table 5: Ablation of result selection strategies for GroupPose [44] with Swin-T.

| Selection Strategy | AP | $AP_M$ | $AP_L$ |
|---|---|---|---|
| None | 33.5 | 30.3 | 36.8 |
| w/ L1 | 47.4 | 45.9 | 48.8 |
| w/ IoU ($\tau$=0.3) | 69.1 | 65.8 | 72.3 |
| w/ Intersection ($\tau$=0.5) | 70.9 | 68.7 | **73.1** |
| w/ Intersection ($\tau$=0.3) | **72.0** | **73.6** | 72.7 |

Table 6: Ablation of multi-task learning.

|  | Text Prompt | | Scribble Prompt | |
|---|---|---|---|---|
|  | Pose | Mask | Pose | Mask |
| w/o Pose Head | - | 61.8 | - | 68.7 |
| w/o Mask Head | 63.3 | - | 72.7 | - |
| **Ours** | **66.7** | **62.2** | **74.7** | **70.0** |

Table 7: Ablation of global dependency modeling.

|  | Text Prompt | | Scribble Prompt | |
|---|---|---|---|---|
|  | Pose | Mask | Pose | Mask |
| w/o Global Dep. | 53.6 | 54.4 | 63.5 | 65.8 |
| Self-Attention | 64.9 | 61.1 | 73.8 | 69.4 |
| **Ours** | **66.7** | **62.2** | **74.7** | **70.0** |

to competitors using higher-resolution inputs, further demonstrating the efficacy of our approach and the significance of the proposed task.

**Comparison with Text-based Segmentation Methods.** To further validate the effectiveness of UniPHD in segmentation, we compare it with popular open-source text-based segmentation methods trained on RefHuman, as shown in Table 4. Using only text prompts for training, our model achieves superior performance compared to competitive methods like SgMg [58] and ReLA [42].

**Qualitative Results.** In Figure 4, we present qualitative results of UniPHD in various challenging scenarios such as crowded scenes, occlusions, similar appearances, dim lighting, and viewpoint changes. UniPHD effectively captures appearance, location, action, and context information to generate high-quality outputs. More visualizations are supplied in the Appendix.

## 5.2 Ablation Study

**Result Selection Strategies.** Table 5 shows the performance of a recent human pose estimation model [44] using various result selection strategies on RefHuman `val`. Without carefully designed strategies, the model struggles to accurately predict target individuals. The distance-based strategy fails to effectively filter out high-scoring irrelevant results, while the intersection-based strategy delivers the best performance.

**Multi-task Learning.** In Table 6, we evaluate the impact of multi-task learning in our approach on RefHuman `val` using the AP metric. Removing either head adversely affects the performance of the other. This validates the effectiveness of our decoder, which enables bidirectional information flow between keypoint and instance queries to enhance both predictions.

Table 8: Ablation of query initialization.

|  | Text Prompt | | Scribble Prompt | |
|---|---|---|---|---|
|  | Pose | Mask | Pose | Mask |
| w/o Initialization | 65.5 | 61.0 | 74.0 | 69.6 |
| **Ours** | **66.7** | **62.2** | **74.7** | **70.0** |

Table 9: Training UniPHD with extra data.

| Prompt | Pose Estimation | | Segmentation | |
|---|---|---|---|---|
|  | AP | PCKh@0.5 | oIoU | AP |
| Point | 81.0 | 94.3 | 88.3 | 72.3 |
| Scribble | 81.2 | 94.9 | 89.4 | 73.2 |

**Pose-centric Hierarchical Decoder.** In Table 7, we analyze the impact of global dependency modeling in our decoder. Without global dependencies, keypoint queries struggle to perceive the prompts for predicting target-aware results. Our graph attention clearly outperforms self-attention because it not only models dynamic node relations but also simulates inherent keypoint-to-keypoint, keypoint-to-instance, and instance-to-keypoint relations via soft adjacent matrix.

**Query Initialization.** Similar to recent transformer-based pose estimation methods [44, 87], we use prompt-conditioned query initialization to enrich queries with dynamic spatial priors. Table 8 reveals that omitting these dynamic priors results in a performance decrease of 0.4-1.2% AP. Nonetheless, our model maintains robust performance without query initialization, demonstrating its efficacy.

**Training with Extra Data.** To unleash the capability of UniPHD for positional prompt-based prediction, we expand our training data beyond RefHuman to encompass the entire MS COCO `train2017` images by generating additional point and scribble prompt annotations based on Bézier curves. As shown in Table 9, UniPHD, trained exclusively with positional prompts on the expanded dataset, achieves impressive performance on RefHuman `val`, with up to **94.9** PCKh@0.5 and **89.4** oIoU, demonstrating its scalability.

# 6 Conclusion

We introduced Referring Human Pose and Mask Estimation in the wild, a new task aimed at simultaneously predicting the poses and masks of specified individuals using natural, user-friendly text or positional prompts. This task holds significant potential for enhancing human-AI interactions in fields such as assistive robotics and sports analysis. To achieve this, we introduced the RefHuman dataset that substantially extends MS COCO with additional text and positional prompt annotations. We also proposed the first end-to-end promptable approach named UniPHD, which employs pose-centric hierarchical decoder to model local details and global dependencies for R-HPM. UniPHD achieves top-tier performance, establishing a solid benchmark for future advancements in this field. We hope our new task, dataset, and approach could foster advancements in human-AI interaction and related areas.

**Broader Impacts.** Malicious use of R-HPM models can bring potential negative societal impacts, such as unauthorized mass surveillance. We believe that the model itself is neutral with positive human-centric applications, including assistive robotics and sports analysis.

**Limitations.** The proposed UniPHD approach produces high-quality results with various prompts. However, text prompts perform worse than positional prompts due to misidentification. Future research could enhance vision-language alignment to reduce this performance disparity.

**Acknowledgments.** This work was supported by the Australian Research Council Industrial Transformation Research Hub IH180100002. Professor Ajmal Mian is the recipient of an Australian Research Council Future Fellowship Award (project number FT210100268) funded by the Australian Government.

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

# A Appendix

## A.1 Training Loss Functions

The overall loss function for UniPHD consists of four parts:

$$\mathcal{L}_{train} = \mathcal{L}_{box} + \mathcal{L}_{class} + \mathcal{L}_{pose} + \mathcal{L}_{mask}, \tag{5}$$

where $\mathcal{L}_{box}$ is for human box regression that contains L1 loss and GIOU [68] loss, $\mathcal{L}_{class}$ is for classification includes focal loss [39] with $\alpha = 0.25$ and $\gamma = 2$, $\mathcal{L}_{pose}$ is for keypoint regression and visibility prediction that includes L1 loss, the constrained L1 loss-OKS loss [70], and cross entropy loss, $\mathcal{L}_{mask}$ is for segmentation that includes Dice [36] loss and focal loss. The loss coefficients $\lambda_{box}^{L1}$, $\lambda_{box}^{GIOU}$, $\lambda_{class}^{focal}$, $\lambda_{pose}^{L1}$, $\lambda_{pose}^{OKS}$, $\lambda_{pose}^{CE}$, $\lambda_{mask}^{Dice}$, $\lambda_{mask}^{focal}$ are 5, 2, 2, 10, 4, 4, 5, 2, following [44, 58, 87]. The Hungarian algorithm [32] is used to identify the optimal assignment (query group) with the highest similarity to the ground truth for training. The class label for the optimal instance query with the minimum loss from the ground truth is set to one, while all others are set to zero.

## A.2 Implementation Details

We follow the common optimization strategies in [44, 58, 83, 87] to train the models. During training, we augment input images through random flip, random crop, and random resize with the shorter sides within the range of 360 to 640 pixels and the longer sides up to 640 pixels. For each iteration, we randomly select either a text or positional prompt with equal probability. We use the AdamW [48] optimizer with a weight decay of $1\times10^{-4}$ and train our models on 24GB RTX 3090 GPUs with batch size 16 for 20 epochs. The initial learning rates are set to $1\times10^{-5}$ for the visual encoder and $1\times10^{-4}$ for other components, with a rate decay at the 18th epoch by a factor of 10. Both the multimodal encoder and pose-centric hierarchical decoder consist of 6 layers, and we use 20 query groups for our models. During testing, we resize the input images with their longer sides up to 640 pixels.

## A.3 Different Query Group Numbers

In Table 10, our approach perform well across different numbers of query groups, measured by the AP metric. This hyperparameter primarily affects pose estimation, with 20 query groups identifying more candidates to achieve the best performance.

Table 10: Ablation of different number of query groups.

| Num. | Text Prompt | | Scribble Prompt | |
|---|---|---|---|---|
| | Pose | Mask | Pose | Mask |
| 5 | 62.6 | 60.1 | 72.1 | 69.1 |
| 10 | 64.4 | 61.4 | 72.4 | 69.5 |
| 20 | 66.7 | 62.2 | 74.7 | 70.0 |

## A.4 Increasing Model Capacity

In Table 11, we enhance model capacity by adding multimodal encoder layers and increasing the feature dimensions from 256 to 384. The results indicate our current settings are already highly effective.

Table 11: Ablation of increasing model capacity.

| | Text Prompt | | Scribble Prompt | |
|---|---|---|---|---|
| | Pose | Mask | Pose | Mask |
| Baseline | **66.7** | 62.2 | 74.7 | 70.0 |
| w/ more layers | 66.3 | 62.0 | **74.9** | **70.4** |
| w/ higher dimensions | 66.6 | **62.8** | 74.5 | 70.2 |

## A.5 Additional Visualizations

In Figure 5, we present additional qualitative results of UniPHD with text and positional prompts. UniPHD effectively processes these prompts to produce high-quality results.

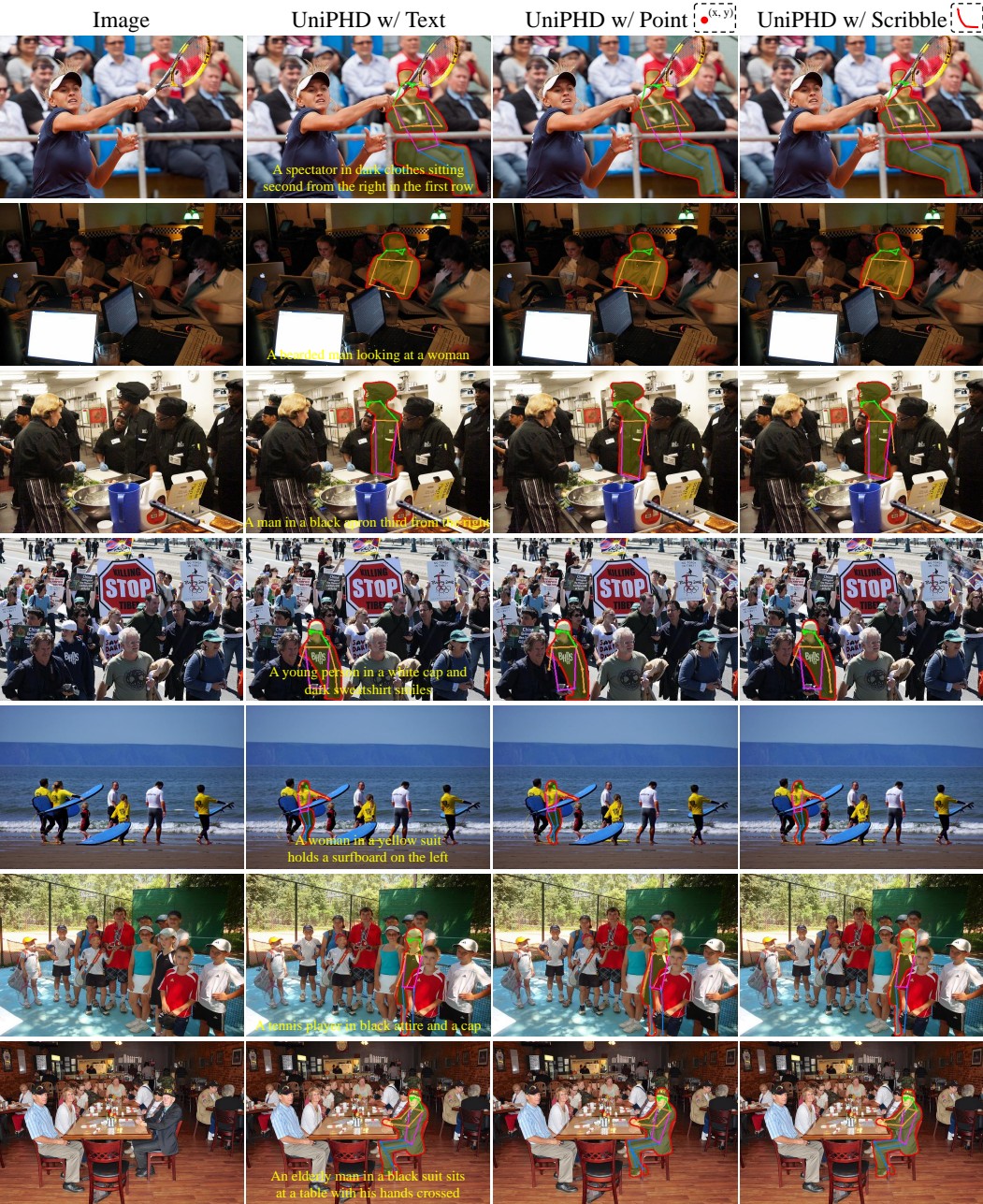

Figure 5: Qualitative results of our UniPHD with different prompts in various challenging scenarios.

**Licenses of MS COCO and RefCOCO/+/g.** The annotations in MS COCO belong to the COCO Consortium and are licensed under CC BY 4.0. The use of the images complies with Flickr Terms of Use. See https://cocodataset.org/#termsofuse for more details. The RefCOCO/+ datasets are licensed under Apache License 2.0 and the RefCOCOg dataset is licensed under CC BY 4.0.

**Terms of Use and License of RefHuman.** RefHuman is licensed under CC BY 4.0.

