# OpenReview forum: "Referring Human Pose and Mask Estimation In the Wild"
_NeurIPS.cc/2024/Conference — NeurIPS 2024 poster_

### Official Review · Reviewer_DJJG · 2024-06-29

**Soundness:** 3
**Presentation:** 3
**Contribution:** 3
**Rating:** 6
**Confidence:** 5

**Summary:**

This paper introduces a new task named as Referring Human Pose and Mask Estimation (R-HPM), which adopt text/point/scribble to represent a specific person and estimate its pose and segmentation mask. To achieve this goal, this paper proposes a new R-HPM dataset named RefHuman and a new method UniPHD to perform R-HPM. Experiments on the proposed dataset and MSCOCO demonstrate the effectiveness of the proposed method.

**Strengths:**

1.	The proposed R-HPM task is useful and complementary to existing HPM tasks, which can bring new insights into this area.
2.	The proposed dataset RefHuman is large and can support the following research in R-HPM. The provided text/point/scribble annotation is complete and can accurately describe a specific person.
3.	Experiments on the proposed dataset and MSCOCO demonstrate the effectiveness of the proposed method in R-HPM task.

**Weaknesses:**

1.	The definition of point and scribble in Sec.3.1 is not clear. Does point prompt only contains only one point and scribble contains 12 points? How to define the point in point prompt? This paper should give a form definition, not textual description.
2.	Some confusion in experiments. First, what is the meaning of dagger in Table 2. If I understand correct, dagger denotes to adopt all images from MSCOCO (~60K) into training, but RefHuman only contains 20K images, how UniHPD can utilize the rest 40K images without ref annotation? Second, I think * should be the default evaluation configuration (namely choose the top-1), so what is the evaluation configuration without * and why adopt it as default configuration?
3.	How about adopting multiple types of prompts to perform R-HPM? For example, using both text and point simultaneously to refer a specific person.
4.	Evaluation metric is not suitable to R-HPM. This paper adopts AP to evaluate performance, which is designed to evaluate multiple objects in one image. However, in referring setting, only single instance is involved, therefore single instance evaluation metric such as PCKh@0.5 is more suitable for R-HPM, or just evaluating the keypoint error/segmentation IoU is ok.
5.	To establish a comprehensive benchmark, the authors are expected to test some existing methods so that the following work can take as reference. None of the compared methods in Table 2 are referring-based methods, so this paper should reimplement some referring-based methods to test their performance on RefHuman, e.g.,  some referring segmentation methods.

**Questions:**

See Weakness.

Overall, I think the motivation and contribution of this work is pretty good. But there are still some questions that should be clarified in the revised paper.

**Limitations:**

This paper already discusses limitation in Sec.6.

---

> ### Author Rebuttal · Authors · 2024-08-06
>
> We sincerely thank Reviewer DJJG for their comments and appreciation of our work. In response to the concerns expressed in Weaknesses and Questions, we provide the following answers:
>
> > Does a point prompt contain only one point, while a scribble contains 12 points? This paper should provide a formal definition for both point and scribble.
>
> Yes, the point prompt contains only one point, while the scribble prompt contains 12 points uniformly sampled from the curve. We will add the following formal definitions as suggested:
>
> **Point prompt**: A single point $\mathbf{p}$=$(x,y)$ at any position in the target area, where $x$ and $y$ are the horizontal and vertical coordinates.
>
> **Scribble prompt**: A scribble can be a continuous, free-form curve represented by an ordered set of $n$ points {$(x_{1},y_{1}), (x_{2},y_{2}), ..., (x_{n},y_{n})$} anywhere in the target area. In this work, we discretize the curve by uniformly sampling 12 points to form the scribble prompt $\mathbf{s}$ = \{$(x_{\left\lfloor kn/12 \right\rfloor},y_{\left\lfloor kn/12 \right\rfloor}) \mid k = 1, 2, \ldots, 12$\}.
>
> > Dagger denotes to adopt all images from MSCOCO into training? How UniHPD can utilize the rest 40K MS COCO images without ref annotation?
>
> Yes, dagger means that all images from MS COCO have been used in training, consistent with previous pose estimation works. Besides the images in RefHuman, we generated point and scribble prompt annotations for the remaining MS COCO images to support the training of UniPHD$^{\dagger}$. We will release these annotations as well.
>
> >  I think * (namely choose the top-1) should be the default evaluation configuration. What is the evaluation configuration without * and why adopt it as default configuration?
>
> We agree with the reviewer and will adopt $^*$ (top-1 ranked query group) as the default evaluation configuration in the final paper, as it reflects practical deployment where a model outputs a single result per input (L289). Some results are shown in Table A, our method still demonstrates top-tier performance under the suggested PCKh\@0.5 and oIoU metrics.
>
> Our method without $^*$ uses 20 query groups for Pose AP and Mask AP evaluation (oIoU always uses only top-1), consistent with previous query-based pose estimation methods. For instance, GroupPose and ED-Pose use over 100 queries to enhance recall, despite most MSCOCO images contain fewer than three people.
>
> *Table A: Comparison on RefHuman using the top-1 query group* ($^*$).
> | | Pose PCKh\@0.5 (Text) | Mask oIoU (Text) | Pose PCKh\@0.5 (Scribble) | Mask oIoU (Scribble)  |
> |:-|:-:|:-:| :-:|:-:|
> | Uni-ED-Pose | 78.3  | 74.5 | 89.6  | 84.9  |
> | Uni-GroupPose  | 78.0 | 74.7 | 89.3  | 85.6  |
> | Ours  | **79.2** | **75.3** | **90.3** | **86.0** |
>
> > How about adopting multiple types of prompts to perform R-HPM? For example, using both text and point simultaneously to refer a specific person.
>
> Interesting idea! We sequentially cross-attend visual features with point and text prompts to enhance multimodal representations for R-HPM, achieving significantly improved results due to the complementary information from both prompts, as shown in Table B. Future research could use our introduced RefHuman dataset to develop more advanced models.
>
> *Table B: Ablation of using both text and point prompts.*
> | Prompts | Pose PCKh\@0.5 | Mask oIoU  |
> |:-|:-:|:-:|
> | Text | 79.2 | 75.3 |
> | Point | 88.7 | 82.5 |
> | Text+Point | **91.4** | **86.6** |
>
> > PCKh\@0.5 and IoU are more suitable as R-HPM metrics.
>
> Thank you for this insightful advice. We have reported the overall IoU for segmentation evaluation, consistent with the referring segmentation task, and will adopt PCKh\@0.5 as the primary pose estimation metric in the revised paper. Please kindly refer to Table A in the previous response. Our model still demonstrates top-tier performance under PCKh\@0.5 and oIoU metrics.
>
>
> > Re-implement some referring segmentation methods to test their performance on RefHuman.
>
> Thanks for the suggestion. We trained popular open-sourced referring segmentation models on our dataset, with results in Table C. Using only text prompt to train our model, we achieve top-tier performance compared to competitors like SgMg and GRES. These results will be included in the revised paper.
>
> *Table C: Comparison to language-conditioned segmentation models.*
> | Method | Backbone | oIoU  |
> |:-|:-:|:-:|
> | LAVT [1] | Swin-T | 74.5 |
> | CGFormer [2] | Swin-T | 75.3 |
> | GRES [3] | Swin-T | 75.9 |
> | SgMg [4] | Swin-T | 75.9 |
> | Ours | Swin-T | **76.3** |
>
> [1] LAVT: Language-Aware Vision Transformer for Referring Image Segmentation, CVPR, 2022.
>
> [2] Contrastive Grouping with Transformer for Referring Image Segmentation, CVPR, 2023.
>
> [3] GRES: Generalized Referring Expression Segmentation, CVPR, 2023.
>
> [4] Spectrum-guided Multi-granularity Referring Video Object Segmentation, ICCV, 2023.

---

> ### Author Response · Authors · 2024-08-13
>
> Dear Reviewer DJJG,
>
> Thank you for your diligent review of our submission. We have carefully addressed each of your concerns and provided our responses. We would greatly appreciate any additional comments, as your feedback is crucial in strengthening our work.
>
> Your time and consideration are invaluable to us.

---

### Official Review · Reviewer_apkP · 2024-07-11

**Soundness:** 3
**Presentation:** 3
**Contribution:** 4
**Rating:** 6
**Confidence:** 4

**Summary:**

In this paper the authors tackle the problem of in-the-wild human pose estimation in a “referring” setting where the goal is to determine the pose of a person referred to using either a text prompt or positional prompt. To achieve this, the authors annotate MS COCO dataset with over 50K annotated instances for 2D keypoints, mask  and prompt (either as text, points or scribbles). They use this dataset to train a model called UniPHD which consists of several submodules. The results show the paper is able to train a strong baseline model which will be useful for future research.

**Strengths:**

S1. This paper introduces a new task of referring human pose estimation by releasing a large dataset of 50K annotations (as an extension to MS COCO) enabling researchers to train models that can interact with the model using text and points/scribble.

S2. The paper also releases a baseline method for the same task with an aim to learn end-to-end R-HPM.

**Weaknesses:**

W1. The motivation for this obtaining pose in a “referring” manner is unclear. For instance, it might be more valuable to focus a text based human detection as opposed to pose estimation because the former can open up many avenues such as human tracking for research.

W2. Multimodal encoder is shallow potentially limiting the interaction of the different encoded features. It might be helpful increase the capacity of the the multimodal encoder model.

W3. Parts of the paper not are fully clear. I understand that the paper aims to be “single-stage” but on L205-L214, there seem to be multiple stages where candidates are detected and then used for subsequent pose estimation. (Which makes it two-stage?)

Minor comments

* Table. 1 does not require last column
* Table. 3 typo “Scibble" —> “Scribble”

**Questions:**

Q1. How does Eq. 4 fit into mask prediction task?  Is that applied only to the pose prediction branch?

**Limitations:**

The authors have briefly acknowledged limitations of their method.

---

> ### Author Rebuttal · Authors · 2024-08-06
>
> We sincerely thank Reviewer apkP for their comments and appreciation of our work. In response to the concerns expressed in Weaknesses and Questions, we provide the following answers:
>
> > The motivation for obtaining pose in a "referring" manner is unclear. It might be more valuable to focus on text-based human detection, which can open up many avenues.
>
> Our task predicts both **mask** and pose of the referred person simultaneously using common prompts to provide detailed, identity-aware human representations that benefit human-AI interaction.
> This has the following advantages over text-based human detection:
>
> (1) Our segmentation provides finer-grained, less noisy information than detection, crucial for robotics and precision tasks.
>
> (2) Our pose estimation adds essential semantics for behavior understanding, complementing mask/box information.
>
> (3) Support for diverse prompts (text, point, scribble), broaden the scope of human-AI interaction.
>
> (4) Our proposed dataset supports various settings, including text-based human detection mentioned by the reviewer.
>
> Reviewer DJJG also thinks the motivation of this work is pretty good.
>
>
> > It might be helpful to increase its capacity.
>
> Thanks for this valuable suggestion. We enhanced the capacity of the multimodal encoder by adding three layers or increasing feature dimensions from 256 to 384. As shown in Table A, both strategies slightly improved overall performance, demonstrating that our current settings are highly effective. We will incorporate the results in the revised paper.
>
>
> *Table A: Increasing the capacity of the multimodal encoder.*
> | | Pose AP (Text) | Mask AP (Text) | Pose AP (Scribble) | Mask AP (Scribble) |
> |:-|:-:|:-:| :-:|:-:|
> | Baseline | **66.6** | 62.1 | 74.6 | 70.0 |
> | w/ more layers | 66.3 | 62.0 | **74.9** | **70.4** |
> | w/ higher dimensions | **66.6** | **62.8** |74.5 | 70.2 |
>
>
> > The method seems to be two-stage, according to L205-214.
>
> To address the reviewer's concern, we will remove the term "one-stage" since it does not affect our contribution. Previous papers like ED-Pose described "two-stage" methods that first perform detection, then pose estimation on cropped single-human images, or use heuristic grouping to process numerous detected instance-agnostic keypoints.
> Our method uses linear layers to identify a high-scoring point $\mathbf{c}$ on multimodal features for initializing keypoint queries (L205-214), which directly regress target keypoint positions on the *entire* image.
> Our query initialization is similar to the human query selection in ED-Pose, which claims to be one stage.
> Hence, we call our method one stage because it is end-to-end and avoids cropping, pose estimation on cropped single-human images, or heuristic grouping.
>
> Moreover, the initialization on L205-214 does not even play a significant role in our method, evidenced by an ablation study using only learnable keypoint queries. As shown in Table B, this results in only a minor performance decrease, demonstrating the effectiveness of our method. We will include this ablation study in the paper.
>
> *Table B: Ablation of query initialization.*
> | | Pose AP (Text) | Mask AP (Text) | Pose AP (Scribble) | Mask AP (Scribble) |
> |:-|:-:|:-:| :-:|:-:|
> | w/o initialization | 65.5 | 61.0 | 74.0 | 69.6 |
> | Ours | **66.6** | **62.1** | **74.6** | **70.0** |
>
> >  Table 1 does not require last column and typo “Scibble" —> “Scribble”.
>
> Thank you for pointing this out. We will remove the last column from Table 1, correct the pointed typo and carefully check the complete paper.
>
>
> > How does Eq. 4 fit into mask prediction task? Is that applied only to the pose prediction branch?
>
>
> Our graph attention, including edge construction (Eq. 4), is applied concurrently to both mask and pose prediction branches. Our query set $\mathbf{Q}^{IP} \in{\mathbb{R} ^{(k+1) \times D}}$ comprises one instance (prompt) query and $k$=17 keypoint queries. The graph attention treats all queries in $\mathbf{Q}^{IP}$ as nodes and models keypoint-to-keypoint, keypoint-to-instance, and instance-to-keypoint relations, enhancing all queries simultaneously. Ultimately, the instance query generates dynamic filters for mask prediction, while the keypoint queries estimate keypoint positions.

---

> ### Author Response · Authors · 2024-08-13
>
> Dear Reviewer apkP,
>
> Thank you for your diligent review of our submission. We have carefully addressed each of your concerns and provided our responses. We would greatly appreciate any additional comments, as your feedback is crucial in strengthening our work.
>
> Your time and consideration are invaluable to us.

---

> > ### Comment · Reviewer_apkP · 2024-08-13
> >
> > The rebuttal has addressed some of the concerns. Hence, I increase the score to a weak accept. I continue to think that there is room to improve the presentation and soundness of the paper, but the contributions are valuable.

---

> > > ### Author Response · Authors · 2024-08-14
> > >
> > > Dear Reviewer apkP,
> > >
> > > We greatly appreciate your recognition of our work's contributions and the upgraded score. Your valuable comments have been crucial in refining our paper. As suggested, we will further improve the presentation and soundness in the final version.

---

### Official Review · Reviewer_TbW6 · 2024-07-11

**Soundness:** 3
**Presentation:** 3
**Contribution:** 3
**Rating:** 5
**Confidence:** 4

**Summary:**

This paper proposes a new task called Referring Human Pose and Mask Estimation and introduces the corresponding RefHuman dataset, which is beneficial for research on human behavior comprehension. Additionally, the authors present a model that leverages three types of prompts for this task. The proposed UniPHD model achieves promising performance in this area.

**Strengths:**

1. The proposed R-HPM task and the RefHuman dataset are beneficial to related research.

2. The proposed method shows promising performance on both the new task and traditional human pose estimation tasks.

**Weaknesses:**

1. The descriptions of the method, especially in Section 4.3, is confusing. For example, what is the relationship between F^(vl) and P’? How do you enhance the template based on P’? And why do “the keypoint queries struggle to perceive the prompt information and lacks interactions with each other, challenging the target-awareness and instance coherence” ? It seems that these keypoint queries can interact with each other in existing decoders. More detailed descriptions and explanations are needed to understand this work.

2. The author proposed that existing research overlooks joint human pose and mask estimation, which provides comprehensive human representations. However, the paper lacks comparisons of the proposed model with other variants, such as UniPHD without the mask head and UniPHD without the pose head. Such comparisons could help clarify the relationship and benefit of the pose estimation and mask estimation tasks.

3. The proposed method needs further validation. Providing an ablation study of the proposed query initialization method and comparing model parameters and computational complexity would be helpful.

**Questions:**

See the weaknesses

**Limitations:**

Yes

---

> ### Author Rebuttal · Authors · 2024-08-06
>
> We sincerely thank Reviewer TbW6 for their comments and appreciation of our work. In response to the concerns expressed in Weaknesses and Questions, we provide the following answers:
>
> > What is the relationship between $\mathbf{F}^{vl}$ and $\mathbf{P}^{'}$, and how do you enhance the template based on $\mathbf{P}^{'}$?
>
> Thank you for the comments. We extract the multimodal embedding $\mathbf{F}^{vl}_{\mathbf{c}}$ at the highest-scoring position $\mathbf{c}$ (evaluated using linear layers) to estimate a set of $k$=17 keypoint positions $\mathbf{P}^{'}$ for better query initialization. We enhance the keypoint query template $\mathbf{Q}^{P} \in{\mathbb{R} ^{k \times D}}$ by adding semantically rich multimodal embeddings $\mathbf{F}^{vl} _{\mathbf{P}^{'}}$ extracted at positions $\mathbf{P}^{'}$, which also serve as reference points for deformable attention in the decoder. We will explain this in more detail in the revised paper.
>
> > Why do "the keypoint queries struggle to ....."? It seems that these keypoint queries can interact with each other in existing decoders.
>
> Perhaps there is some misunderstanding which can be removed by rephrasing the sentence as "However, *after local detail aggregation,* the keypoint queries struggle to perceive .......".
> We first update each query embedding separately by capturing local details, which lacks query interactions. Therefore, we mention this in L222 to introduce our global dependency modeling, which captures keypoint-to-keypoint, keypoint-to-instance, and instance-to-keypoint relations to enhance target awareness and instance coherence. We did not claim that other works' decoders cannot perform keypoint query interactions.
>
> >  Comparisons of UniPHD with its variants, such as UniPHD without the mask head and UniPHD without the
> pose head.
>
> Thanks for the valuable suggestion. We performed the comparisons in Table A, which reveal that removing either head adversely affects the performance. This confirms the effectiveness of our synergistic decoder, which facilitates keypoint-instance interactions, enabling bidirectional information flow and enhancing both predictions. We will add this ablation study to the paper.
>
> *Table A: Ablation of prediction heads.*
> | | Pose AP (Text) | Mask AP (Text) | Pose AP (Scribble) | Mask AP (Scribble) |
> |:-|:-:|:-:| :-:|:-:|
> | w/o pose head | - | 61.8 |  - | 68.7 |
> | w/o mask head | 63.3 | - | 72.7 | - |
> | Ours | **66.6** | **62.1** | **74.6** | **70.0** |
>
> > Provide an ablation study of the proposed query initialization.
>
> Table B shows the ablation study on query initialization by removing the query enhancement discussed in our response to the first comment. Results show a minor performance decrease due to the absence of dynamic spatial priors.
>
> *Table B: Ablation of query initialization.*
> | | Pose AP (Text) | Mask AP (Text) | Pose AP (Scribble) | Mask AP (Scribble) |
> |:-|:-:|:-:| :-:|:-:|
> | w/o enhancement | 65.5 | 61.0 |  74.0 | 69.6 |
> | Ours | **66.6** | **62.1** | **74.6** | **70.0** |
>
>
> > Compare model parameters and computational complexity.
>
> Thank you for this suggestion. Table C compares the model parameters and FPS (measured on a single RTX4090 GPU). Our method demonstrates better performance at comparable FPS while using slightly more parameters. We will include this comparison in the revised paper.
>
> *Table C: Comparison with competitors using the scribble prompt for inference.*
> | | Pose AP | Mask AP | FPS | Params |
> |:-|:-:|:-:| :-:|:-:|
> | Uni-ED-Pose | 72.9 | 69.2 | 50 | **175.7M** |
> | Uni-GroupPose | 73.0 | 69.0 | **52** | 177.7M |
> | Ours | **74.6** | **70.0** |  50 | 184.0M |

---

> ### Author Response · Authors · 2024-08-13
>
> Dear Reviewer TbW6,
>
> Thank you for your diligent review of our submission. We have carefully addressed each of your concerns and provided our responses. We would greatly appreciate any additional comments, as your feedback is crucial in strengthening our work.
>
> Your time and consideration are invaluable to us.

---

> > ### Comment · Reviewer_TbW6 · 2024-08-13
> > **Post-rebuttal**
> >
> > I appreciate the authors' response. The additional explanations and ablation studies improve the quality of the paper and effectively demonstrate the design's effectiveness. As a result, I have raised the score.

---

> > > ### Author Response · Authors · 2024-08-13
> > >
> > > Dear Reviewer TbW6,
> > >
> > > Thank you for your positive feedback! We are glad that our rebuttal has addressed your concerns. Your constructive comments have been very helpful in refining our work, and we will incorporate these additional results in the final paper.

---

### Author Rebuttal · Authors · 2024-08-06

We sincerely appreciate the reviewers for their thoughtful and constructive feedback.

We are glad that **all** the reviewers recognize the importance of our newly proposed task of Referring Human Pose and Mask Estimation and appreciate the significance of our new dataset, RefHuman. The reviewers also agree that the proposed UniPHD method achieves top-tier performance on the RefHuman and MS COCO datasets.

We have conducted all additional ablations and experiments suggested by the reviewers. We will revise our manuscript according to all the comments and remain committed to continuous improvement. We eagerly await your final decision.

---

### Author Response · Authors · 2024-08-13

Dear Reviewers,

Thank you for your diligent effort in reviewing our submission! We have carefully addressed the concerns raised and conducted the requested experiments. We would greatly appreciate any additional comments and are ready to engage in further discussion. If there are no further issues, we respectfully request that you consider revising the score for our paper in light of the improvements we have made.

Thank you again for your time and consideration.

---

### Decision · Program_Chairs · 2024-09-25

**Decision:**

Accept (poster)

**Comment:**

Following the rebuttal, this work received positive feedback, with two weak accepts and one borderline accept. The authors effectively addressed most of the reviewers' queries, leading two reviewers to raise their scores. The contribution of the paper, particularly the introduction of the R-HPM task and the RefHuman dataset—a substantial extension to MS COCO with 50K annotations—was well appreciated. This dataset is expected to enable further research in related research area of human-AI interactions. The authors are encouraged to incorporate the suggested ablations and enhance the presentation in line with the reviewers' comments.